

# Expression of leptin and leptin receptors in colorectal cancer—an immunohistochemical study

Saad M. Al-Shibli[1,*], Norra Harun[2], Abdelkader E. Ashour[1], Mohd Hanif B. Mohd Kasmuri[3] and Shaikh Mizan[1,*]

[1] Department of Basic Medical Sciences, International Islamic University, Kuantan, Pahang, Malaysia
[2] Pathology Department, Hospital Tengku Ampuan Afzan, Kuantan, Pahang, Malaysia
[3] Department of Pathology & Laboratory Medicine, International Islamic University, Kuantan, Pahang, Malaysia
[*] These authors contributed equally to this work.

## ABSTRACT

Obesity is demonstrated to be a risk factor in the development of cancers of various organs, such as colon, prostate, pancreas and so on. Leptine (LEP) is the most renowned of the adipokines. As a hormone, it mediates its effect through leptin receptor (LEPR), which is widely expressed in various tissues including colon mucosa. In this study, we have investigated the degree of expression of LEP and LEPR in colorectal cancer (CRC). We collected 44 surgically resected colon cancer tissues along with normal adjacent colon tissue (NACT) from a sample of CRC patients from the Malaysian population and looked for leptin and leptin receptors using immunohistochemistry (IHC). All the samples showed low presence of both LEP and LEPR in NACT, while both LEP and LEPR were present at high intensity in the cancerous tissues with 100% and 97.7% prevalence, respectively. Both were sparsed in the cytoplasm and were concentrated beneath the cell membrane. However, we did not find any significant correlation between their expression and pathological parameters like grade, tumor size, and lymph node involvement. Our study further emphasizes the possible causal role of LEP and LEPR with CRC, and also the prospect of using LEPR as a possible therapeutic target.

## INTRODUCTION

Adiposity is considered a major health problem of pandemic dimension and involves both developed and developing countries (*WHO, 2016*; *Popkin, Adair & Ng, 2012*; *Misra & Khurana, 2011*). Much surpassing its physical and mechanical burden, it comes out as a significant metabolic player in various endocrine and cardiovascular disorders, and has also been demonstrated to be a risk factor in the development of cancers of various organs, such as colon, esophagus, gall bladder, pancreas, kidney, thyroid, prostate, uterus and breast (*WHO, 2016*; *Bhaskaran et al., 2014*; *NCI, 2016*).

Colorectal cancer (CRC) is one of the most common causes of cancer-related death worldwide (*Yu et al., 2014*). The epidemiological evidence shows a clear link between

Corresponding author
Shaikh Mizan,
shaikhmizan2015@gmail.com

colon cancer and obesity (*Robsahm et al., 2013*; *Harriss et al., 2009*; *Potter, 1999*). The two are associated with a sedentary lifestyle, high-energy diets, and limited consumption of vegetables, fruits, and fibers (*Robsahm et al., 2013*; *Harriss et al., 2009*; *Potter, 1999*). The relationship between the two, if any, needs to be studied through deeper investigations.

Adipose tissue synthesizes and secretes a number of hormones and cytokines, also called adipokines, such as leptin (LEP), adiponectin, resistin, apelin, omentin, tumor necrosis factor-a, IL-6, etc (*Kuryszko, Sławuta & Sapikowski, 2016*; *Booth et al., 2016*; *Goodwin & Stambolic, 2015*). The tissue acts upon other organs through the adipokines. The most widely researched and most significant obesity-related adipokine is leptin (LEP) (*Friedman & Halaas, 1998*; *Guo et al., 2012*; *Friedman, 2015*; *Friedman & Mantzoros, 2015*; *Lipsey et al., 2016*).

It is established that LEP mediates its action through its receptor (LEPR) (*Schwartz et al., 2000*; *Ha et al., 2013*; *Allison & Myers, 2014*). Therefore, LEPR also has become an important target of research as part of the LEP-LEPR system for humoral control of organs and their functions.

A baffling feature of LEP is that although its most abundant source is white adipose tissue (WAT), it is also secreted by many other tissues; for example, placenta (*Hoggard et al., 1997*; *Masuzaki et al., 1997*), stomach (*Bado et al., 1998*), mammary gland (*Smith-Kirwin et al., 1998*), brain and pituitary, (*Morash et al., 1999*), colon (*Hardwick et al., 2001*), ovaries (*Löffler et al., 2001*), bone and cartilage cells (*Morroni et al., 2004*), testicles (*Soyupek et al., 2005*), skeletal muscle (*Solberg et al., 2005*), and so on.

Likewise, the leptin receptor (LEPR) is found in many tissues other than its canonical target organ: the hypothalamus. The receptor is expressed in placenta (*Amico et al., 1998*; *Ebenbichler et al., 2002*), gastric mucosa (*Mix et al., 2000*; *Sobhani et al., 2000*; *Breidert et al., 1999*), lung (*Tsuchiya et al., 1999*), endometrium (*Kitawaki et al., 2000*), immune cells (*Caldefie-Chezet et al., 2001*), liver (*Otte et al., 2004*), and pancreas (*Tudurí et al., 2013*). Adipose tissue and gastric mucosa also produce forms of LEPR that are soluble in plasma and tissue fluids and remain bound to circulating leptin increasing its half-life (*Cammisotto & Bendayan, 2007*).

Consistent with the wide occurrence of LEP and LEPR, the system besides its canonical function of balancing food intake and body mass (*Friedman & Halaas, 1998*; *Guo et al., 2012*; *Friedman & Mantzoros, 2015*; *Lipsey et al., 2016*; *Schwartz et al., 2000*; *Ha et al., 2013*; *Allison & Myers, 2014*; *Ahima, 2008*; *Bjørbaek & Kahn, 2004*) is found to regulate a plethora of processes and pathways, including growth of and secretion from gastric epithelial cells (*Mix et al., 2000*), secretion of mucin from colonic goblet cells (*Plaisancie et al., 2006*); reproduction, bone remodeling, insulin signaling, neuroendocrine function (*Slattery et al., 2008*) inflammatory response (*Aloulou et al., 2008*; *Shamsuzzaman et al., 2004*), regulation of blood pressure, thyroid hormone release, immune factors and cells (*Bado et al., 1998*; *Barrachina et al., 1997*; *Adeyemi et al., 2005*), pancreatic beta cells, regulation of fat and glucose metabolism, insulin sensitivity, and so on *Bjørbaek & Kahn (2004)*.

The most important action of the LEP-LEPR system in relation to tumorigenesis is its growth effects. It has been found to be required as a growth factor for mammary gland (*Hu et al., 2002*; *Esper et al., 2015*; *Cleary et al., 2004*), fetal and neonatal growth (*Masuzaki*

*et al., 1997*; *Heiman et al., 1997*), lung (*Tsuchiya et al., 1999*), hepatic cells (*Chen et al., 2007*), pancreatic $\beta$ cell growth and function (*Morioka et al., 2007*), colonic epithelial cells (*Hardwick et al., 2001*), and so on. Bone growth and bone mass are severely reduced in LEP deficient (ob/ob) animals, but it can be restored with the administration of LEP (*Steppan et al., 2000*). More significantly, the system can interact with a number of other hormonal mediators including insulin, glucagon, the insulin-like growth factors, estrogen, progesterone, growth hormone and glucocorticoids (*Margetic et al., 2002*). Notably, to execute its growth effects, it has been demonstrated that the system promotes cell proliferation, angiogenesis, mesenchymal transformation, and exerts anti-apoptotic effect (*Lipsey et al., 2016*; *Russo et al., 2004*; *Endo et al., 2011*; *Mencarelli et al., 2011*; *Guo, Liu & Gonzalez-Perez, 2011*; *Mullen & Gonzalez-Perez, 2016*; *Ghasemi et al., 2019*), which also are essential requirements of tumorigenesis (*Guo et al., 2012*; *Mullen & Gonzalez-Perez, 2016*; *Surmacz, 2013*).

As evidence to the above hypothesis, LEP and LEPR have been demonstrated in unusually high concentration in various cancerous tissues by many authors (*Koda et al., 2007a*). They are found in high concentration in breast carcinoma (*O'brien, Welter & Price, 1999*; *Ishikawa, Kitayama & Nagawa, 2004*; *Al-Shibli et al., 2017*), leukemia (*Konopleva et al., 1999*), as well as prostate (*Saglam et al., 2003*), esophagus (*Somasundar et al., 2003*), gastric (*Hong et al., 2006*), lung (*Ribeiro et al., 2006*), adenocarcinomas, etc.

Many authors have reported high presence of LEP in colorectal cancerous cells (*Koda et al., 2007b*; *Paik et al., 2009*; *Liu et al., 2011*; *Wang et al., 2012*; *Yoon et al., 2014*; *Jeong et al., 2015*). Recently, a study in Saudi Arabia on colorectal tumors has found LEP in a very high percentage (93%) of the samples on immunostaining (*Al-Maghrabi, Qureshi & Khabaz, 2018*). Nevertheless, some authors reported that in advanced cancers LEP expression diminishes (*Hong et al., 2006*; *Koda et al., 2007b*), suggesting silencing of LEP expression in an advanced stage, which indicates the anti-tumorigenic role of the LEP. Again, *Aparicio et al. (2005)* have reported that LEP acts as an in vitro growth factor for colon cancer cells, but does not promote tumor growth *in vivo*. Such conflicting reports make the case of LEP in CRC all the more interesting.

Many studies have also demonstrated over-expression of LEPR in colon cancer (*Hardwick et al., 2001*; *Aparicio et al., 2005*; *Hoda et al., 2007*; *Drew, 2012*; *Erkasap et al., 2013*; *Mu et al., 2014*). Altered patterns of LEPR expression have been reported by a number of authors (*Aloulou et al., 2008*; *Uchiyama et al., 2011*; *Uddin et al., 2009b*; *Stachowicz et al., 2010*). It has been proposed that phenotypic variation of LEPR expression may give variants with better prognosis (*Aloulou et al., 2008*; *Uddin et al., 2009b*). Some other authors have reported that in CRC patients, tissue LEP and LEPR are related significantly to the grade of tumor differentiation, depth of bowel wall invasion, and distant metastasis (*Joshi & Lee, 2014*). The confounding complexity of LEP-LEPR system in human colon cancer patients obviously demands wider studies.

In this paper, we report an investigation of the degree of expression/presence of both LEP and LEPR using immunohistochemistry in colorectal mucosa in surgically resected CRC specimens from a sample of the Malaysian population.

**Table 1  Age distribution of patients with colorectal carcinoma.**

| Age groups: (years) | 30 s | 40 s | 50 s | 60 s | 70 s | 80 s |
|---|---|---|---|---|---|---|
| $n = 44$ | 4 | 5 | 8 | 19 | 6 | 2 |
| (in %) | (9.1) | (11.4) | (18.2) | (43.2) | (13.6) | (4.5) |

**Table 2  Sex distribution of patients with colorectal carcinoma.**

| Sex: | Female | Male |
|---|---|---|
| $n = 44$ | 18(40.9%) | 26(59.1%) |

**Table 3  Tumor grade distribution of the patients with colorectal carcinoma.**

| Tumor grade | Well differentiated | Moderately differentiated | Undifferentiated |
|---|---|---|---|
| $n = 44$ | 2(4.5%) | 40(91%) | 2(4.5%) |

# MATERIALS & METHODS

## Tissue samples

A total of 44 paraffin blocks of colorectal cancer (CRC) were taken from the histopathology laboratory in Hospital Tengku Ampuan Afzan (HTAA), Kuantan, Pahang, Malaysia. Colon samples were taken from resected tumors along with the adjacent normal tissue, which was used as controls. Selection of patients and clinical diagnosis were done in collaboration with the Department of Histopathology at HTAA from January 2017 to May 2018.

The age distribution of the patients was as such: 4 (9.1%), 5 (11.4%), 8 (18.2%), 19 (43.2%), 6 (13.6%) and 2 (4.5%) were in their 30s, 40s, 50s, 60s, 70s, and 80s respectively. Most of the samples were from males 26 (59.1%). Most 40 (91%) were moderately differentiated, only 2 (4.5%) of samples were well differentiated and only 2 (4.5%) undifferentiated. Age, sex and tumor grade distribution of the patients are shown in Tables 1–3.

## Tissue collection and preparation

The study protocol will include studying of histopathological samples from 44 patients with colorectal cancer. From the patients' forty-four pairs of histopathological samples, each pair consisting of a sample from the cancer tissue and another from adjacent normal colon tissue (ANCT) were obtained by the histopathological laboratory (HTAA). The tissue samples taken from ANCT of the same patient were considered as controls. All tissue samples were subjected to histopathological examination using immunohistochemistry procedure detailed below.

A rotary microtome machine was used for sectioning. Trimming and sectioning were done with about 4–5 μm thick. All slides were stained with H&E and stored until the histopathological examination was achieved.

All histopathological procedures were conducted at the research laboratories in the Department of Basic Medical Sciences, Faculty of Medicine, International Islamic University Malaysia (IIUM), Kuantan, Pahang, Malaysia in collaboration with the Department of Pathology at HTAA.

## Immunohistochemistry (IHC)

IHC for Leptin and Leptin receptors were done using a Dako Autostainer (Dako, Glostrup, Denmark), utilizing the REAL$^{TM}$ EnVision$^{TM}$ Detection System, Peroxidase/Dab+, Rabbit protocols from Dako, Denmark. Tissue sections were made on saline coated glass slides for IHC staining. AB9749 Abcam Rabbit polyclonal to Leptin and AB104403 Abcam Rabbit polyclonal to Leptin Receptor were used.

Anti-leptin antibody and anti-leptin receptor antibody were diluted to 1:1000 and 1:100 respectively by adding the appropriate amount of Dako antibody diluent.

IHC staining was performed using a Dako Autostainer according to the manufacturer's protocol. The IHC-stained tissue sections were counterstained in hematoxylin solution for 15 s and were rinsed in running tap water for 5 min before differentiating in 1% acid alcohol for 2 dips. The sections were then rinsed in running tap water for 5 min. The tissue sections were dehydrated in 70% alcohol for 3 min, followed by 80%, 90% and absolute alcohol 2 and 1.5 min respectively. Then, the tissue sections were dried in an oven for 10 min at 37 °C, followed by clearing in xylene twice for 2 min. The slides were finally cover-slipped with DPX mounting medium.

IHC stained slides were examined under low and high power magnification using an Olympus BX15 (Tokyo, Japan) light microscope. The result was then analyzed for the expression of leptin and Leptin receptors.

In our study, the immunoreactivity of Leptin and leptin receptor has been examined, by two pathologists, for staining intensity and positively stained cells percentage. The frequency of positive cells was evaluated by applying a semiquantitative method. Staining intensity has been given scores 0, 1, 2, and 3 demonstrating negative, faint, moderate, and strong staining respectively. For simplification of analysis, scores of staining intensity have been grouped as negative, low (1+) and high (2+ and 3+), as was reported previously (*Koda et al., 2007b*; *Al-Maghrabi, Qureshi & Khabaz, 2018*).

## Statistical analysis

SPSS software (version 22.0) was used for all statistical calculations. Spearman's rho coefficient was used to determine correlations between the variables.

## Ethical approvals

The full research protocol was approved by the International Islamic University Malaysia Research Ethical Committee (IREC). The approval reference number is IIUM/305/20/4/1/7. Since already recorded data and stored surgical tissue samples were issued no consent was thought necessary or feasible.

**Table 4   Percentage of positive (low and high) cases of LEP and LEPR in IHC stained of colorectal tissue.**

|  | Immunoreactivity for LEP | | Immunoreactivity for LEPR | |
|---|---|---|---|---|
|  | Low | High | Low | High |
| Cancer tissue $N = 44$ | 0 | 44 (100%) | 1(2.3%) | 43 (97.7%) |
| Adjacent normal colon tissue $N = 44$ | 44 (100%) | 0 | 44 (100%) | 0 |
| P value* |  | <0.01 |  | <0.01 |

Notes.

*P values for significant difference in expression of LEP and LEPR between NACT and cancerous colon tissue were calculated by applying Wilcoxon signed rank test.

IHC, immunihistochemistry; LEP, leptin; LEPR, leptin receptor; NACT, normal adjacent colon tissues.

**Table 5   Results of the statistical analysis of the correlation of LEP and LEPR with the common pathological parameters.**

|  | Tumor size | Lymph node involvement | Grade (W,M,U)* |
|---|---|---|---|
| LEP correlation coefficient (Spearman's rho, P value) | $-0.054, P = 0.73$ | $-0.205, P = 0.181$ | $-0.000, P = 1.00$ |
| LEPR correlation coefficient (Spearman's rho, P value) | $-0.018, P = 0.907$ | $-0.206, P = 0.179$ | $-0.285, P = 0.06$ |

Notes.

*W: Well differentiated cells; M: Moderately differentiated cells; U: Undifferentiated cells.

LEP, leptin; LEPR, leptin receptor.

# RESULTS

## Presence of leptin (LEP) and leptin receptor (LEPR) in the cancerous and the normal adjacent colon tissue samples

We selected 44 out of 56 CRC tissue samples as qualitatively acceptable for the purpose of our study. Among the 44 CRC samples, moderately differentiated (M) tumors were significantly higher compared to well differentiated (W) and undifferentiated (U) samples, 40, two and two, respectively ($P < 0.01$, Chi-square value = 61.714). Among the cancerous tissues, all of the 44 (100%) samples stained strongly (2+ or 3+) for LEP and all but one or 43(97.7%) samples stained strongly for LEPR with the IHC technique. In contrast, all of the 44 normal adjacent colon tissue (NACT) took low stain for both LEP and LEPR. The result is tabulated in Table 4.

## Results of the statistical analysis

The difference in the expressions of LEP and LEPR between the cancerous and NACT were very significant ($P < 0.01$, Wilcoxon signed rank test). The statistical values are summarized in Table 5.

However, we did not find any significant correlation between their expression and pathological parameters like grade, tumor size, and lymph node involvement (Table 5).

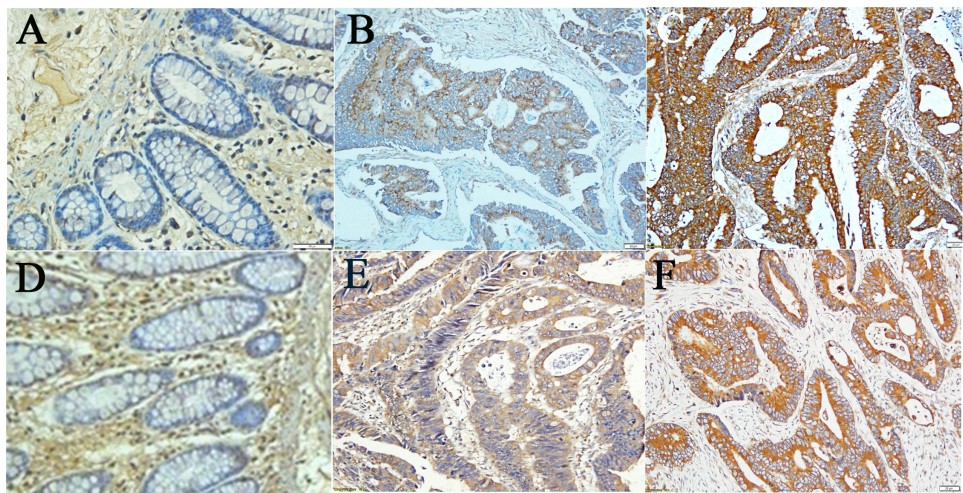

**Figure 1** **Results from Immunohistochemical (IHC) staining.** Immunohistochemical (IHC) staining for LEP (A–C), and LEPR (D–F): (A,D) Adjacent Normal Colon Tissue (ANCT) with low staining (intensity 1+), (B,E) Moderately differentiated CRC with high staining (intensity (2+), (C,F) Moderately differentiated CRC with high staining (intensity 3+); (all ×200 magnification).

## DISCUSSION

Definite association of obesity with various types of tumors readily drew attention towards its flag bearer hormone leptin (LEP), as it is the humoral mediator through which adipose tissue could exert a direct effect on other organs and processes. An indispensable part of LEP action is its receptor (LEPR). In this study, we graded staining as 'negative,' 'low,' and 'high' (as elaborated in methodology) and observed presence of both LEP and LEPR in colorectal cancer (CRC) tissue samples in high intensity with high prevalence, 100% and 97.7% respectively ($p < 0.01$) (Table 4). However, adjacent normal colon tissue (ANCT) were weakly and invariably immunostained for both LEP and LEPR (100%). In contrast to the previous report by *Koda et al. (2007b)*, the distribution of stain in the ANCT was comparatively uniform too (Fig. 1).

We do not know of any other report so far that has found such a high occurrence of LEP and LEPR in colonic or any other types of tumors. The numerically nearest prevalence of LEP (not LEPR) has been reported by *Al-Maghrabi, Qureshi & Khabaz (2018)*, where LEP has been observed to be present in 93.5% of the CRC cases in the Western Province of Saudi Arabia, but moderate to strong staining in only 22.75% of the cases. While findings of *Al-Maghrabi, Qureshi & Khabaz (2018)* match with those of *Jeong et al. (2015)*; previous reports by *Koda et al. (2007b)* from Europe, *Paik et al. (2009)* from Korea, *Liu et al. (2011)* and *Wang et al. (2012)* from China showed further lower values (LEP was just present only in 51.2%, 73.5%, 72.1, and 71.3%, respectively). Although the presence of both LEP and LEPR in CRC tissues in this Malaysian sample of the population is strikingly high and interesting, it would be premature to draw any conclusion about whether the differences are due to regional, temporal or racial factors, before further independent studies.

*Koda et al. (2007b)* have reported that LEP is overexpressed in human CRCs relative to normal colorectal mucosa. While in normal mucosa it is low or undetectable, its level is higher in tissues adjacent to CRCs. But LEP concentration is the highest in moderately differentiated (G2) cancers compared to poorly differentiated (G3) cancers. Similarly, *Al-Maghrabi, Qureshi & Khabaz (2018)* reported that larger size tumors gave a significantly higher proportion of negative immunostaining. Lower LEP occurrence has also been reported with less differentiated gastric adenocarcinomas, and the authors suggested silencing of LEP/LEPR expression in advanced stages of cancers (*Hong et al., 2006*). It is possible that in advanced stages of cancer strong oncogenes takes over the processes and the LEP system is overwhelmed at least in some cases. However, we did not find any negative correlation of LEP with advanced CRC. We had only two undifferentiated cases out of 44, but both the undifferentiated samples immunostained strongly for LEP. However, the number of cases of undifferentiated CRCs are statistically inadequate in our study to draw any conclusion. We did not find any significant correlation of LEP or LEPR staining with tumor grade. Our samples also failed to show any significant correlation either with lymph node involvement or invasion (Table 5). These findings match with those of *Al-Maghrabi, Qureshi & Khabaz (2018)* and *Jeong et al. (2015)*, however, they contrast with *Koda et al. (2007b)* and *Liu et al. (2011)*.

Concomitant occurrence of high concentration LEP and LEPR in various cancerous cells are so far explained by co-expression of both in the same cell (*Ishikawa, Kitayama & Nagawa, 2004*; *Jardé et al., 2008*; *Garofalo et al., 2006*). However, an alternative explanation might be that overexpression of only LEPR is followed by binding and trapping of the equivalent amount of LEP from circulation. Such a situation would result in overstaining of both in the respective cells. This hypothesis is supported by the findings of *Stachowicz et al. (2010)*, who did not find mRNA for LEP in human CRC tissue samples. Interestingly, *Erkasap et al. (2013)* reported over-expression of LEPR mRNA in human metastatic CRCs, but not in CRCs of local origin. On the other hand, some authors reported expression of LEP mRNA in normal colonic cells (*Attoub et al., 2000*). These controversies demand much more detailed study of the expression of LEP and LEPR through mRNA studies instead of protein staining.

The very high presence of LEPR, as is found in our study, must be very significant as it is the receptor of LEP. The presence of only high level of circulating LEP cannot be sufficient to produce excessive growth promotion leading to cancer; it must need the receptor. The over-expression of LEPR, as we found in one of our previous studies with breast cancer (*Al-Shibli et al., 2017*), and by many authors with various cancers (*Aloulou et al., 2008*; *Ishikawa, Kitayama & Nagawa, 2004*; *Mu et al., 2014*; *Uchiyama et al., 2011*; *Uddin et al., 2009b*; *Jardé et al., 2008*; *Miyoshi et al., 2006*; *Uddin et al., 2009a*; *Hoon Kim et al., 2008*) must be significant in the carcinogenic influence of obesity or LEP. However, the claims by various authors about the effect of LEP-LEPR system in colon cancer are confounding, and maybe even contradictory (*Hoda et al., 2007*; *Uchiyama et al., 2011*; *Stattin et al., 2003*; *Stattin et al., 2004*; *Bolukbas et al., 2004*; *Kumor et al., 2009*; *Sălăgeanu et al., 2010*; *Kosova et al., 2013*). Some authors reported increased LEP levels in male but not in female CRC patients (*Stattin et al., 2003*; *Stattin et al., 2004*). Other showed that LEP expression in

cancer tissue rises as carcinogenesis progresses (*Koda et al., 2007b*; *Paik et al., 2009*) and that LEP expression in cancer tissue may be positively correlated with survival of colorectal cancer patients (*Paik et al., 2009*) or LEPR over-expression is associated with anti-tumor response and better prognosis (*Aloulou et al., 2008*); while some claim that serum LEP values are lower in patients with colon cancer (*Bolukbas et al., 2004*; *Kumor et al., 2009*), serum LEP levels decreases with the progress and aggressiveness of tumor (*Sălăgeanu et al., 2010*); while others even fail to determine any significant difference in serum LEP between colon cancer patients and controls (*Soyupek et al., 2005*). An explanation to such contradictory report might lie in the fact that LEPR shows polymorphism and phenotypic variants (*Aloulou et al., 2008*; *Mu et al., 2014*; *Uchiyama et al., 2011*; *Uddin et al., 2009b*) and it is very much possible that different variants respond differently to LEP stimulation—one variant might over-stimulate cells towards carcinogenesis, while another might be less sensitive to circulating LEP concentration or even act against the progress of the tumor.

## CONCLUSIONS

Over-expression of both LEP and LEPR in CRC, along with similar findings in our previous studies with breast carcinoma and related works of other authors, as mentioned above, suggest strong association of LEP/LEPR system with carcinogenesis. Given the already established role of LEP/LEPR system in growth promotion, it may be assumed to be causally related to carcinogenesis, where its growth effect may go out of control and result in tumorous growths or its hyperactivity may help the tumor to grow faster.

Since LEP/LEPR is over-expressed in so many processes, they could not be generally used as cancer markers; however, if specific phenotypic variants of LEPR are involved with cancers, which is yet to be studied, then such phenotypes could be used as cancer markers. Apparent contradictions about the role of LEPR may be due to phenotypic variants of LEPR, which may respond differently to LEP stimulation. It seems that while the LEPR pathway has much potential to be a target for anti-cancer drug therapy, drugs might be better targeted to particular variants of LEPR. Therefore, elaborate characterizations of phenotypic variants of LEPR are essential for various reasons.

## ACKNOWLEDGEMENTS

Our thankfulness would be directed to all the faculties, friends and staffs in the Department Basic Medical Sciences, Faculty of Medicine, International Islamic University Malaysia; and to the Department of Pathology, Hospital Tengku Ampuan Afzan, Kuantan, Pahang, Malaysia for their extraordinary assistance.

### Funding

This work was supported by International Islamic University Malaysia. The funders had no role in study design, data collection and analysis, decision to publish, or preparation of the manuscript.

### Grant Disclosures

The following grant information was disclosed by the authors:
International Islamic University Malaysia.

### Competing Interests

The authors declare there are no competing interests.

### Author Contributions

- Saad M. Al-Shibli conceived and designed the experiments, performed the experiments, analyzed the data, contributed reagents/materials/analysis tools, prepared figures and/or tables, authored or reviewed drafts of the paper, approved the final draft, coordinated between labs.
- Norra Harun conceived and designed the experiments, performed the experiments, analyzed the data, contributed reagents/materials/analysis tools, prepared figures and/or tables, approved the final draft.
- Abdelkader E. Ashour conceived and designed the experiments, analyzed the data, prepared figures and/or tables, authored or reviewed drafts of the paper, approved the final draft.
- Mohd Hanif B. Mohd Kasmuri performed the experiments, contributed reagents/materials/analysis tools, approved the final draft, did a lot of bench work.
- Shaikh Mizan conceived and designed the experiments, performed the experiments, analyzed the data, prepared figures and/or tables, authored or reviewed drafts of the paper, approved the final draft.

### Human Ethics

The following information was supplied relating to ethical approvals (i.e., approving body and any reference numbers):

The full research protocol was approved by the International Islamic University Malaysia Research Ethical Committee (IREC; approval reference number: IIUM/ 305/20/4/1/7.).

### Data Availability

The raw data is available as a Supplemental File.

### Supplemental Information

Supplemental information for this article can be found online at http://dx.doi.org/10.7717/peerj.7624#supplemental-information.

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
