# Peer review of "Expression of leptin and leptin receptors in colorectal cancer—an immunohistochemical study"

_PeerJ, doi:10.7717/peerj.7624_

## Round 0.1 · original submission · Minor Revisions

Dear Dr. Mizan,

Thank you for submitting your manuscript entitled, " Expression of leptin and leptin receptors in colorectal cancer - an immunohistochemical study," to PeerJ. All the reviewers had very positive comments about the work submitted for peer review. Despite, the positive comments two reviewers have raised few minor technical and conceptual concerns which unfortunately preclude publication of the current version of the manuscript in PeerJ. The revised manuscript in response to the review critiques will likely be accepted for publications. Please submit a copy of the revised manuscript with "tracked" or highlighted changes, as well as an unmarked or "clean" version.

We appreciate the opportunity to review your work and look forward to the revised version of the manuscript.

Sincerely,

Sarwat Naz, PhD
Academic Editor

·

Basic reporting

Professional English was used throughout the manuscript.
The literature references were detailed.
The Tables were of professional quality. The Figure quality was good.
Note: I was unable to find any Figure legend in the manuscript.
The results were relevant and supported the hypothesis.

Experimental design

The experimental design was relevant to the study area. The statistical analysis was detailed.

Validity of the findings

The results were elaborate and the discussion was well written.

Additional comments

Manuscript title: Expression of leptin and leptin receptors in colorectal cancer – an immuno-histochemical study
Shibli, SM et al. have performed immuno-histochemical analysis on 44 colorectal cancer patients. There results demonstrated that both Leptin and Leptin receptor are over-expressed in surgically resected colon cancer tissues when compared to normal adjacent colon tissue.
The manuscript is well written. The results and conclusions are crisp and to the point. The discussion section was compiled in a very professional manner. I would like to compliment the authors for comparing their findings with previously reported literature in great details. In my opinion, the manuscript is suitable for publication.
Some minor comments:
1) Figure legend for Figure 1 was missing.
2) A comparison of LEP and LEPR levels in cancerous tissues versus inflamed colon tissues (from non-cancerous patients) would have made this manuscript even more interesting. This would have helped in understanding if LEP and LEPR can be used as a potential marker to differentiate between colorectal cancer and non-cancerous colorectal inflammation related diseases.
3) Please consider removing lines 31 and 32 where the authors mention that they are the 1st to report such a high presence of LEP and LEPR in colorectal cancer in any part of the world. Reason: IHC results are based on the techniques and reagents used across the samples. Moreover, grading in IHC results (eg. +1. +2 or +3), until quantified by specific digital methods, is not comparable among multiple studies, unless the same pathologist grades the tumors at the same time.
4) Please reconsider modifying lines 280 and 281. Based on this manuscript, the increase in LEP and/or LEPR expression has not yet been characterized to be a cause or an effect of neoplastic progression. Moreover, the role of LEP and LEPR signaling pathway has not been characterized in this manuscript. Thus, the authors may reconsider their conclusion of considering the LEPR pathway as a potential candidate for anti-cancer therapy. Alternatively, the authors can include that their findings combined with research from some other research groups (as mentioned in the introduction part of this manuscript) make the LEPR pathway an interesting candidate for potential anti-cancer therapy.

Reviewer 2 ·

Basic reporting

Well written English and literature citation.

Experimental design

Methods and research question well defined.

Validity of the findings

Conclusions need revision to bridge with old and future findings.

Additional comments

Comments on manuscript # 37549 "Expression of leptin and leptin receptors in colorectal cancer - an immunohistochemical study. " by Al-Shibli et. al.

The author studied the expression status of hormones and hormone receptor (Leptin and Leptin receptor) in a bunch of colorectal cancer tissues as well as the adjacent normal tissues but the role is still missing as per author’s findings.

Following things need to be addressed in the MS.
1. Line 93-95 and line 264-266: Normal adjacent tissues showed low expression of LEP and LEPR, but CRC tissues showed high expression of LEP and LEPR and author claimed in accordance with other authors findings that high expression is correlated with high prognosis, how? That explanation is missing in MS, which is the main finding of this MS.
2. Line 80 and 97: LEP (ob/ob) animals showed less bone growth/mass but increase on LEP administration and LEP acts as a growth factor in-vitro and not in-vivo-these are itself controversial statements. Should be explained in MS?
3. Line 235: Author stated that patch distribution of LEP was found in CRC tissues in accordance with reference 80. Is that the purity reason of the tissue, if author claimed that it is not differentiation specific or grade specific problem? Please explain in MS.
4. Figure 1: H&E staining should be included to see the proliferation/tissue integrity of the tissues used for IHC staining?
5. Validation of IHC results with other technique, like Western blotting or qPCR-would confirm the handling/analysis of IHC results.
6. Conclusion of the findings of this MS w.r.t. novelty or addition to the already known findings is specifically lacking.
The paper is well written in nice English. Comments above- will make the MS better.

---

## Round 0.2 · accepted · Accept

Dear Dr. Mizan,

Thank you for your submission to PeerJ.

I am writing to inform you that your revised manuscript - Expression of leptin and leptin receptors in colorectal cancer - an immunohistochemical study - has been Accepted for publication. Congratulations!

·

Basic reporting

Professional English was used throughout the manuscript.
The literature references were detailed.
The Tables were of professional quality. The Figure quality was good.
The results were relevant and supported the hypothesis.

Experimental design

The experimental design was relevant to the study area. The statistical analysis was detailed.

Validity of the findings

The results were elaborate and the discussion was well written.

Additional comments

Manuscript title: Expression of leptin and leptin receptors in colorectal cancer – an immuno-histochemical study
Shibli, SM et al. have performed immuno-histochemical analysis on 44 colorectal cancer patients. There results demonstrated that both Leptin and Leptin receptor are over-expressed in surgically resected colon cancer tissues when compared to normal adjacent colon tissue.
The manuscript is well written. The results and conclusions are crisp and to the point. The discussion section was compiled in a very professional manner. I would like to compliment the authors for comparing their findings with previously reported literature in great details.
The authors have made the necessary modifications to the manuscript as suggested in my 1st review.
In my opinion, the manuscript is now suitable for publication in PeerJ.

Reviewer 2 ·

Basic reporting

Well written in Professional English and literature citation.

Experimental design

Research question well defined and performed.

Validity of the findings

Results nicely concluded as per data.

Additional comments

Author revised MS as per all reviewer's comments, including mine and now the revised MS looks clear and understandable as per scientific point of view.